# How do international humanitarian aid workers stay healthy in the face of adversity?

**Kaz De Jong**[1]*, **Saara Martinmäki**[2], **Hans Te Brake**[2], **Rolf Kleber**[3,4], **Joris Haagen**[2], **Ivan Komproe**[5]

**1** Médecins Sans Frontières, Paris, France, **2** ARQ Centre of Expertise for the Impact of Disasters and Crises, Diemen, Netherlands, **3** Utrecht University, Utrecht, Netherlands, **4** ARQ National Psychotrauma Centre, Diemen, Netherlands, **5** Healthnet TPO, Amsterdam, Netherlands

* Kaz.de.Jong@amsterdam.msf.org

## Abstract

### Background

International humanitarian aid workers (iHAWs) are motivated strongly to travel abroad to help communities affected by war, famine, disaster and disease. They expose themselves to dangers and hardships during their field assignments. Despite working under such challenging circumstances, most workers remain healthy. The objective of the present study was to unravel the mechanism that enables workers to remain healthy under the same circumstances that affect these communities. We hypothesised that the different components of the Sense of Coherence (SOC) health mechanism mediate the relationship between field stressors and post-assignment health.

### Methods and findings

The stress-health model was tested among 465 international aid workers using a longitudinal pre-post assignment study design and structural equation modelling for path analyses. The (health) outcome variables were PTSD, anxiety, depression, emotional exhaustion, and work engagement. Our findings highlight the importance of being healthy before aid assignment and the negative health impact of field stressors that were not potentially traumatic. The SOC components mediated the relationship between field stress and post-assignment health, with males and females using different SOC components to alleviate stress. Males are more likely trying to understand the nature of the stressor, whereas females mobilise their resources to manage stressors. In both groups, a high level of meaningfulness of the work was an important component in maintaining health. Regarding using the SOC concept for understanding the process of maintaining health, our findings indicated that SOC is best considered context-specific and multidimensional.

### Conclusion

In addition to good pre-mission health, the SOC components help prevent field assignment-related negative health effects in iHAWs. Our findings support the idea to compose gender-balanced teams of iHAWs to maintain and promote health. The findings can be used to

**Data Availability Statement:** Data cannot be shared publicly because of conflicts with the requirement of protecting participants' privacy and confidentiality. Data are available from the MSF

Institutional Data Access / Ethics Committee (contact via MSF for researchers who meet the criteria for access to confidential data: Bern-Thomas Nyang'wa (bern.nyangwa@london.msf.org). The data underlying the results presented in the study are also available from the corresponding author, Kaz.de.jong@amsterdam.msf.org.

**Funding:** This research was supported by MSF Operational Centre Amsterdam. The funder provided financial support in the form of a grant that also included salaries for authors (KdJ, SM, HtB, RJK, JFGH, IK). The funder did not have any additional role in the study design, data collection and analysis, decision to publish, or preparation of the manuscript. The specific roles of these authors are articulated in the 'author contributions' section.

**Competing interests:** Dr. Kaz de Jong is employed by Medecins Sans Frontiers (MSF) the research funder. To ensure no potential conflicts could arise, the data collection and statistical analyses were performed by independent researchers without involvement of dr. de Jong or any other employee of MSF. The results were interpreted with the full research team, including independent researchers and external supervisors (university professors, RK, IK) to ensure quality and independence. MSF also signed a written agreement that it would adhere to the code of conduct for scientific integrity of the Royal Netherlands Academy of Arts and Sciences regarding this research. Both the funding of MSF as well as a clear description of the role of the financing body is mentioned in a separate statement in the manuscript. The role of the different authors in the research is described in detail in the manuscript.

develop or refine health conversation tools and SOC based health interventions to promote health and wellbeing and prevent ill-health among aid workers and other stress-exposed populations.

## Introduction

Approximately 40.000 international humanitarian aid workers (iHAWs) provide urgent humanitarian care to people in need all over the world. These trained professionals make a deliberate, conscious choice to work in extremely stressful and demanding settings of war, natural disaster, or pandemics with overwhelming medical needs of large numbers of affected individuals. IHAWs' work environments are often poorly structured, highly insecure, and exposure to potentially traumatic events (e.g., violence, extreme suffering) is likely. Their workload is high with limited job autonomy and poor remuneration. Personal stressors such as poor work-life balance, minimal privacy after work (living in teams), and being away for long periods from their social network are further examples of the unique challenges iHAWs' are confronted with.

Despite these challenges, most iHAWs maintain their pre-assignment level of health and work engagement [1]. Rather than looking for risk factors in a population with low prevalence to optimize iHAWs' health another perspective may be useful [2]. This triggers the central question of this manuscript: How do iHAWs stay healthy in the face of unsolvable and unavoidable adversity? What prevents or mitigates the impact of the stressful environment and promotes health and work engagement?

The Salutogenetic Model of Health [3] is a frequently used theory to explain how individuals maintain and regain their health while engaging in highly demanding work and overwhelming life challenges. Salutogenesis refers to the ability to manage stress in behavioural, cognitive, and motivational ways.

The mechanism to achieve Salutogenesis is Sense of Coherence (SOC). SOC is defined as 'a global orientation that expresses the extent to which one has a pervasive and enduring though dynamic, feeling of confidence that one's internal (within the person; emotions and thoughts) and external environments (between persons, social experiences, events) are predictable. As well as the belief that there is a high probability that things will work out as well as can reasonably be expected' ([3], p. 123). SOC consists of three key components. First, the cognitive ability to clarify and structure the nature of stressors (comprehensibility). Second, being aware of one's available resources and keeping confidence to mobilise them via behavioural and instrumental responses to manage stressors successfully (manageability). Third, the willingness and motivation to manage stressors depending on whether it makes sense to mobilize resources to deal with these challenges (meaningfulness) [4].

Unravelling the working of SOC on iHAWs' health and work engagement can provide the key to designing preventive activities and interventions resulting in staff health improvements. In many different stressful and demanding circumstances high SOC levels were associated with good health and well-being [5, 6]; low SOC levels with higher levels of disease and mortality incidence [7]. SOC mediated the relationship between stressors and health or well-being [8]. A high SOC decreases the negative effects of field stressors; it acts as go-between stressors and good health. Contrary to national staff humanitarian workers [9] the mediating role of SOC has not been examined in the population of iHAWs.

The SOC component 'meaningfulness' may be a leading mechanism in the iHAWs' process of staying healthy. The act of 'doing good' and the advocacy of deploying personal presence,

capacities and skills in a context of oppression or neglect are strong internal motivators for iHAWs [10, 11]. IHAWs mentioned moral reasons most frequently as a motivation for deployment in the West African Ebola outbreak [12]. A high level of 'meaningfulness' ensures the willingness and motivation of iHAWs to face the extraordinary levels of stress associated with humanitarian work. At the same time when 'giving meaning' to one's actions fails, it was associated with poor mental health (anxiety, depression, burn-out) [13–15].

Sex differences moderated the effects of SOC on health in an elderly community sample [16]. In mental health sex differences in prevalence, symptomatology, and influencing factors are both fascinating and poorly researched [17]. We considered important potential sex differences in salutogenetic mechanisms and pathways to improve our understanding and tailor health-improving interventions to both women and men.

The present study aims to determine whether the SOC components act as mediators between humanitarian aid work-related stressors and iHAWs' health and well-being. We hypothesized that pre-field assignment SOC, especially the component meaningfulness mediates (weakens) the relationship between field stressors, work engagement and post-assignment health. We also hypothesized that the three SOC components act differently among males and females.

## Method

### Participants and procedure

The current study was a prospective survey of 465 iHAWs of Médecins Sans Frontières Operational Centre Amsterdam (MSF OCA). This sample is a subset of the larger sample previously described in detail elsewhere [1]. All iHAWs who completed at least one of the health outcome variables at both pre-and post-assignment was included in the present study. Sixty-one per cent of the participants were females. The average participant age at the start of their participation was 40.5 years old (SD = 11.0, range 24.4–76.5). Independent non-MSF researchers informed all iHAWs going to a field assignment about the study between December 2017 and February 2019; data collection ended February 2020. Participants signed informed consent and completed questionnaires on an online survey platform. Pre-assignment measurement (T1) took place 0–14 days before departing to the assignment area, the second measurement (T2) immediately post-assignment, at a maximum of four weeks after returning. This study received ethical approval from the internal Ethics Review Board of Médecins Sans Frontières on the 24th of February 2017 (ID 1642).

### Instruments

**Health and work engagement outcome indicators.**   The *Maslach Burnout Inventory*, emotional exhaustion scale (MBI-HSS, range: 0–6) [18] measures burnout-related complaints of emotional exhaustion. The internal consistency of this subscale was good (α = .84).

The *Hopkins Symptom Checklist* (HSCL-25, range anxiety and depression: 1–4), assesses symptoms of anxiety and depression during the past week [19]. The internal consistency in the current sample was good for both depression (α = .90) and anxiety (α = .87) subscales.

The *Post-Traumatic Check List DSM-5* (PCL-5, range: 0–80) [20] measures the DSM-5 symptoms of PTSD. In the current sample, the scale had good internal consistency (α = .89).

*Utrecht Work Engagement Scale* (UWES-9, range 0–6) measures work engagement. The definition of high work engagement is 'positive, fulfilling, work-related state of mind' [21]. The internal consistency was good (α = .84).

**Health mechanism indicator.**   *Sense of Coherence* (SOC, range: 13–91) measures three components: comprehensibility, manageability, and meaningfulness [4]. People with a high

SOC are able to manage (extreme) stressors in keeping their good health. The internal consistency of the scale in the current sample was good (α = .80), to modest for the subscales (comprehensibility α = .65, manageability α = .59, and meaningfulness α = .58). The reduction in alpha is likely attributable to the small number of items that comprise each subscale [22]. The mean inter-item correlation provides a more suitable reliability test because it is not hampered by the limited number of items of the subscales [23]. According to Briggs and Cheek, the optimal correlation ranges between .20 and .40. Scores below .20 are likely to measure heterogeneous dimensions instead of a single dimension, whereas scores above .40 are indicative of redundant items. The present study mean inter-item correlations were all within the optimal range (comprehensibility .27, manageability .27, meaningfulness .26).

**Stressor indicators.** The *Life Events Checklist for DSM*-5 (LEC-5 range: happened to me) [24] assesses the prevalence of 16 potentially traumatic events, and one additional item assessing any other extraordinarily stressful event in a respondent's life. We used this measure the amount of exposure to PTE's during the assignment (T2).

*Humanitarian Field Stressor List (HFSL).* The Psychosocial Care Unit (MSF OCA) developed an instrument that measured the severity of 39 potential assignment-related stressors in six dimensions [1] (S2 Appendix). The six dimensions are field conditions, cultural stressors, work-related stressors, team stressors, self-experienced traumatic experiences, and code of conduct. The answers were scored on a six-point scale ranging from 0 ('none/not applicable') to 5 ('high'). A total sum score (range 0–195) was used, with a higher score denoting higher experienced stressor levels. The HFSL is not a screener for a specific latent construct, but was used as an inventory of the number and types of experienced field stressors.

## Statistical analyses

Descriptive analyses were conducted with SPSS (version 23.0) and structural equation modelling (SEM) was applied for path analyses with MPlus (version 8) to test the likelihood of specific pathways between predictors of wellbeing and five different health outcomes. To explore the strength of linear relationships between the variables at different time points, Pearson correlations were calculated between all the health and work engagement variables and predictive membership and stressor variables. T-tests were carried out for each health and work engagement variable to examine whether there were significant differences in the pre- and post-assignment indicators for the whole sample and whether there were significant differences between males and females.

To improve understanding of the temporal relationships between the SOC components and the health outcome variables, the SEM path analyses were directed by pre-defined steps and stages. In the 'pre-modelling' stage, each outcome variable was modelled as a function of its baseline value, baseline SOC components, field stressors, and traumatic stressors. Following that, the post-assignment SOC components were added to the model. We specified a pathway from post-assignment SOC components to the health outcome variable, also included the auto-regression (in time) of the SOC components. In the final pre-modelling step, we treated SOC as a latent construct defined by its three observed indicators (i.e., components). This allowed us to examine whether the temporal relationship of the latent SOC construct was the same as for each SOC component.

After completing the pre-modelling steps, the 'main' mediation models were estimated (see 'hypothesized mediation model' section below). On the basis of modification indices provided by MPlus, we specified several hierarchically nested models, evaluating the model fit at each step. New pathways were specified based on theoretical considerations and added only when they improved the model fit substantially. The final optimized model only contained

significant pathways. This way we derived the most likely best fitting (MLBF) [25, 26] model for each outcome variable in the total sample. Subsequently, the robustness of the MLBF models was tested separately for the male and female subgroups, to see whether the model represented the relationships between the variables in the model equally well for males and females. If the overall model is robust across genders, it is expected the earlier found pathways remain significant. If they do not, they are sensitive for gender differences. We report on the hypothesized mediation models for each health outcome, their optimized MLBF models, and the robustness of the MLFB models for males and females.

There was a minor missing data problem in the analyses concerning all outcome variables besides emotional exhaustion. When a case was missing data on exogenous variables, it was not included in the analyses (1.1–4.1% of the cases in the analyses). To examine and compare how well the various models represented the (co)variations present in the data, the models were evaluated according to maximum likelihood fit indices, including the discrepancy (chi-square), comparative fit index (CFI), root-mean-square error of approximation (RMSEA), and standardized root mean square residual (SRMR). Models that fit well are indicated by CFIs $\geq$ 0.90, RMSEAs $\leq$ 0.08, and SRMRs $\leq$ 0.05 [27]. As our intention was to look for the best model out of several theoretically sound models, the model fit indices were used to guide our selection of the best model out of these theoretically based models.

**Hypothesized mediation model.** Our hypothesized model defined the relationships between field stressors, traumatic stressors, different components of SOC and health outcomes before (T1) and after (T2) humanitarian field assignment (Fig 1). The hypothesis was that pre-assignment health has the strongest associations with post-assignment health, and field stressors and traumatic stressors exert a detrimental effect on the post-assignment health outcome. Based on previous research [1, 28], it was hypothesized that higher scores on SOC components would be associated with more positive health at T2. Besides specifying direct pathways from the traumatic and field stressors and the SOC components to the health outcome, mediation effects of T2 SOC components were also modelled. More specifically, the T2 mediating role of

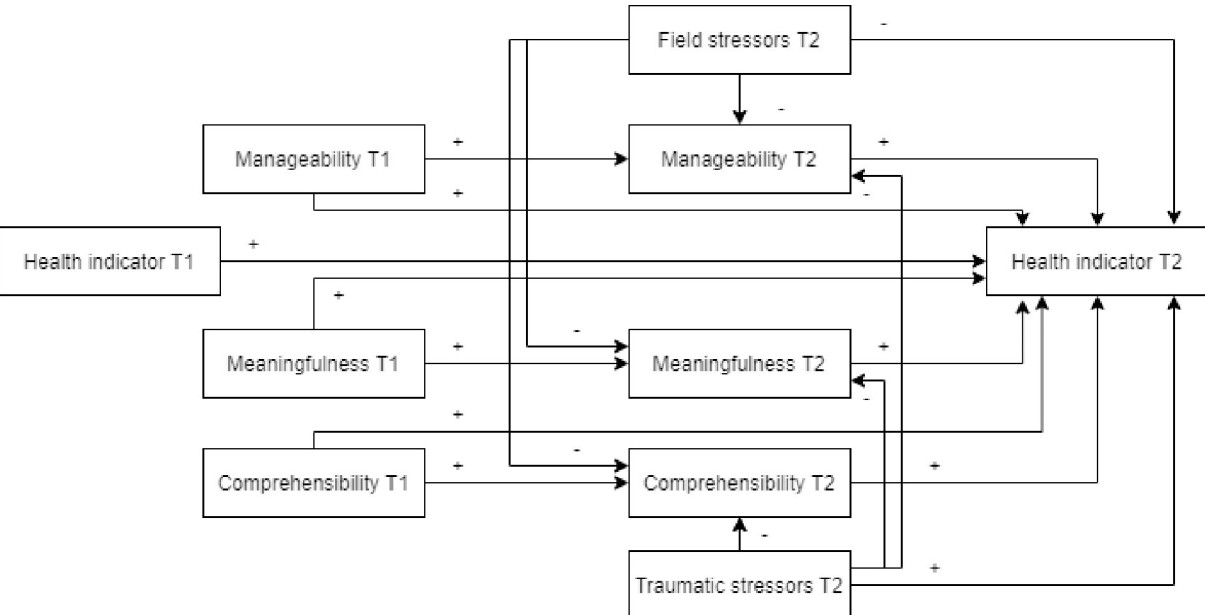

**Fig 1. Hypothesized model about the relationship between the health outcome and its predictors.** A plus sign denotes a positive/beneficial association; a negative sign denotes a detrimental association.

field stressors and traumatic stressors in the relationships between SOC components and the health outcomes were tested. Our assumption was that both stressor types would have a detrimental impact on the SOC components, thereby reducing the so-called 'true protective (beneficial) effect' of the SOC components on health. Based on potential gender differences, the models were tested separately for males and females.

## Results

### Descriptive statistics

The descriptive of the variables chosen for the hypothesized model are presented in Table 1. Compared to men, females reported significantly more symptoms of anxiety, depression, and emotional exhaustion both before and after humanitarian field-assignment. There were no significant differences between males and females in their mean post-traumatic stress symptoms

**Table 1. Descriptive statistics of the health outcome variables before assignment (T1) and after assignment (T2).**

| | | Males | | Females | | Difference |
| --- | --- | --- | --- | --- | --- | --- |
| | Timepoint | Mean | SD | Mean | SD | Stat. test |
| Anxiety | T1 | 1.45 | .03 | 1.55 | .03 | $t(561) = -2.40^*$ |
| | T2 | 1.34 | .03 | 1.44 | .03 | $t(472) = -2.20^*$ |
| | Difference | $t(174) = 3.74^{***}$ | | $t(282) = 3.84^{***}$ | | |
| Depression | T1 | 1.51 | .04 | 1.65 | .03 | $t(544) = -3.46^{**}$ |
| | T2 | 1.49 | .04 | 1.65 | .03 | $t(472) = -2.97^{**}$ |
| | Difference | $t(174) = .32$ | | $t(282) = .21$ | | |
| Emotional exhaustion | T1 | 1.58 | .06 | 1.78 | .06 | $t(587) = -2.26^*$ |
| | T2 | 1.72 | .09 | 1.95 | .07 | $t(479) = -2.12^*$ |
| | Difference | $t(186) = -2.42^*$ | | $t(290) = -3.19^{**}$ | | |
| PTSD | T1 | 8.47 | .58 | 8.99 | .55 | $t(569) = -.10$ |
| | T2 | 7.56 | .58 | 8.43 | .59 | $t(474) = -.88$ |
| | Difference | $t(180) = 1.62$ | | $t(285) = .86$ | | |
| Work engagement | T1 | 4.80 | .06 | 4.70 | .05 | $t(546) = 1.32$ |
| | T2 | 4.73 | .06 | 4.60 | .05 | $t(463) = 1.73$ |
| | Difference | $t(171) = 1.61$ | | $t(270) = 2.59^*$ | | |
| Manageability | T1 | 21.13 | .27 | 20.21 | .21 | $t(590) = 1.66$ |
| | T2 | 21.54 | .27 | 20.19 | .24 | $t(481) = 2.97^{**}$ |
| | Difference | $t(190) = -1.37$ | | $t(290) = -.12$ | | |
| Meaningfulness | T1 | 21.86 | .33 | 21.80 | .21 | $t(590) = -.06$ |
| | T2 | 22.52 | .31 | 22.30 | .21 | $t(481) = .09$ |
| | Difference | $t(190) = -2.42^*$ | | $t(290) = -2.67^{**}$ | | |
| Comprehensibility | T1 | 25.31 | .35 | 24.11 | .28 | $t(590) = 2.47^*$ |
| | T2 | 26.03 | .40 | 24.73 | .29 | $t(482) = 2.51^*$ |
| | Difference | $t(191) = -2.83^{**}$ | | $t(290) = -2.50^*$ | | |
| Field stressors | T2 | 58.27 | 2.12 | 63.64 | 1.67 | $t(466) = -1.61$ |
| Traumatic stressors | T2 | .70 | .10 | .75 | .08 | $t(477) = .23$ |

$^*$ p = < .05

$^{**}$ p = < .01

$^{***}$ p = < .001. Anxiety = HSCL-25 anxiety subscale mean score. Depression = HSCL-25 depression subscale mean score. Emotional exhaustion = MBI-HSS emotional exhaustion subscale mean score. PTSD = PCL-5 total score. Work engagement = UWES-9 mean total score. Manageability = SOC-13 manageability subscale total score. Meaningfulness = SOC-13 meaningfulness subscale total score. Comprehensibility = SOC-13 comprehensibility subscale total score. Field stressors = Humanitarian Field Stressors list total score. Traumatic stressors = LEC-5 number of self-experienced potentially traumatic events.

or work engagement. Post-assignment manageability, as well as pre- and post-assignment comprehensibility, were the only other variables of the model with significant sex differences in the mean scores. In general, males scored higher than females. Males and females did not differ on changes over time (direction and significance) with the exception of the indicator work engagement: there was a general decrease over time, but females decreased significantly. Considering that both males and females decrease in the same direction, the differences between males and females are likely trivial.

## Correlations

S1 Appendix shows the correlation tables between the health and work-engagement outcome variables and the various stressor indicators and the predictive membership indicators. Higher scores on SOC subscales (both pre- and post-assignment) were significantly associated with better health and work engagement at post-assignment. Post-assignment SOC scores had a stronger association with the post-assignment health outcomes than did pre-assignment SOC scores. Higher reported field stressors were significantly associated with higher symptomatology or lower work engagement at post-assignment. Based on the explored correlations between the variables of our interest, we defined the hypothesized path model.

## Pre-modelling

Each outcome variable went through the pre-modelling steps described in the methods. Overall, the models showed that pre-assignment health had a strong positive impact on post-assignment health, that field stressors had a negative impact on some health outcomes, and that while SOC subscales at post-assignment had positive impacts on health outcomes, we could not find the same positive impact between pre-assignment SOC subscales and the health outcomes. The SOC total scale variable demonstrated a comparable pattern of associations within the model as its separate components (manageability, meaningfulness, and comprehensibility). These analyses provide support for all expected pathway associations, except for some contradictory associations for SOC at different time points in relation to the main outcomes, which is likely due to a response shift between the two different time points. Pre-assignment SOC had significant negative associations with all post-assignment health and work engagement indicators, whereas post-assignment SOC also had some significant positive associations (see S1 File).

## Main mediator models

**Anxiety.** *Model.* The hypothesized mediation model (Fig 2, N = 453) did not show an acceptable fit with the variance-covariance matrix of the total sample's data ($\chi2(12) = 314.28$, RMSEA .236, CFI .757, SRMR .116). There were ten significant pathways in this model: as hypothesized, post-assignment anxiety was predicted by its pre-assignment value, by field stressors, and post-assignment manageability and meaningfulness. Congruent with our hypothesis, these two SOC components acted in a protective manner; the higher the SOC component, the lower the post-assignment anxiety. The path from post-assignment comprehensibility to post-assignment anxiety was not significant (p > .05), similar to all paths originating from traumatic stressors. Pre-assignment SOC components were not significantly predictive of post-assignment anxiety. In addition to a direct effect, field stressors also had a significant indirect detrimental effect on post-assignment anxiety.

*Optimized model.* Guided by theoretical considerations and modification indices given by MPlus, the model was optimized by adding covariations between the post-assignment SOC components, adding paths from baseline manageability to post-assignment comprehensibility,

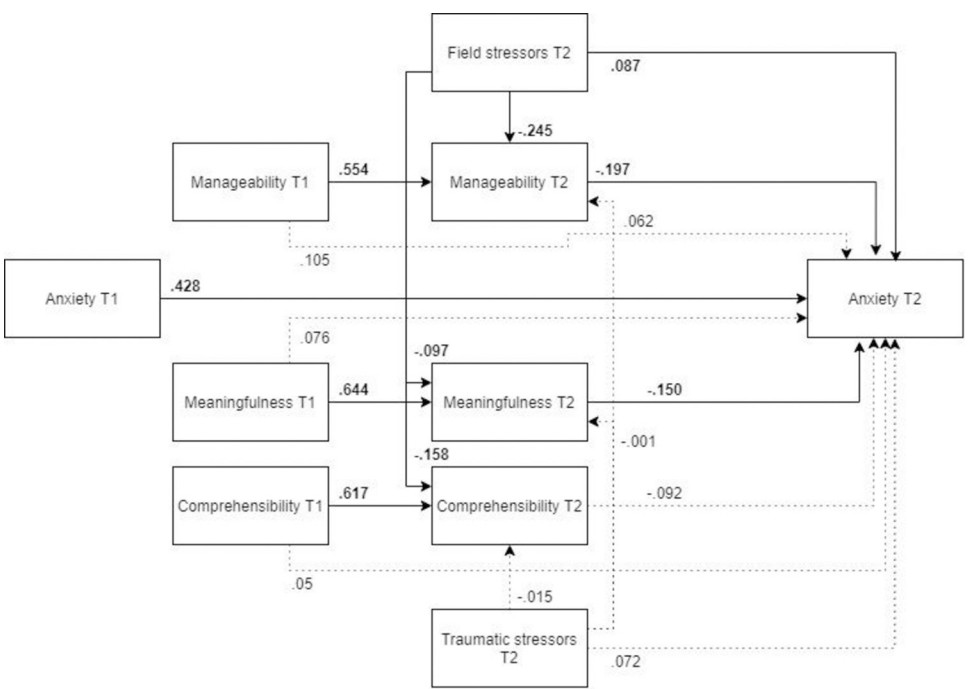

**Fig 2. Tested structural equation model for anxiety.**

as well as from baseline comprehensibility to post-assignment manageability, and removing non-significant paths (Fig 3, N = 453). In this optimized model, pre-assignment manageability was in the end a significant predictor of post-assignment anxiety. All the other remaining pathways were significant, and the MLBF model was a significantly better representation of the data than the original hypothesized model, showing a reasonable fit ($\chi2(9) = 29.566$, RMSEA = .072, CFI = .983, SRMR = .046). Based on our hypothesis that the SOC components may act different among males and females, we proceeded to test the robustness of the optimized model in both subgroups.

*Male and female models.* **Males.** The optimized model was a worse fit for the male subsample than for the female subsample (Fig 4, N = 173). Compared to the optimized model estimated for the whole sample, neither pre- nor post-assignment manageability was a significant

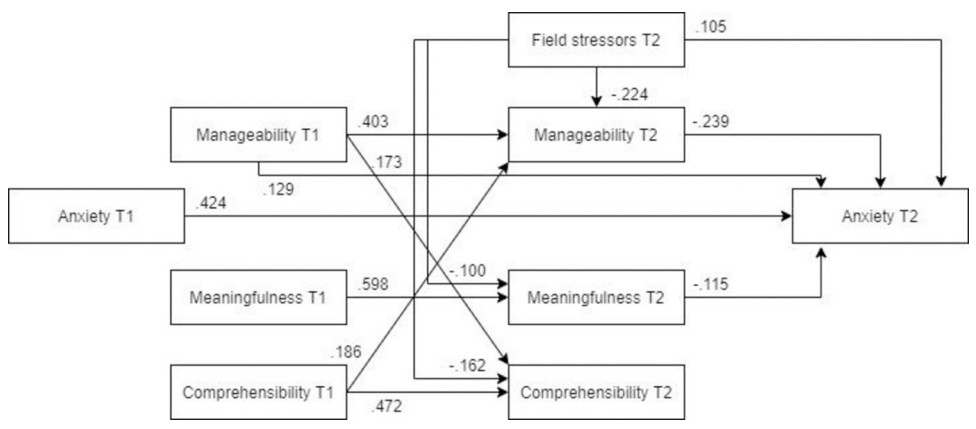

**Fig 3. Optimized structural equation model for anxiety.**

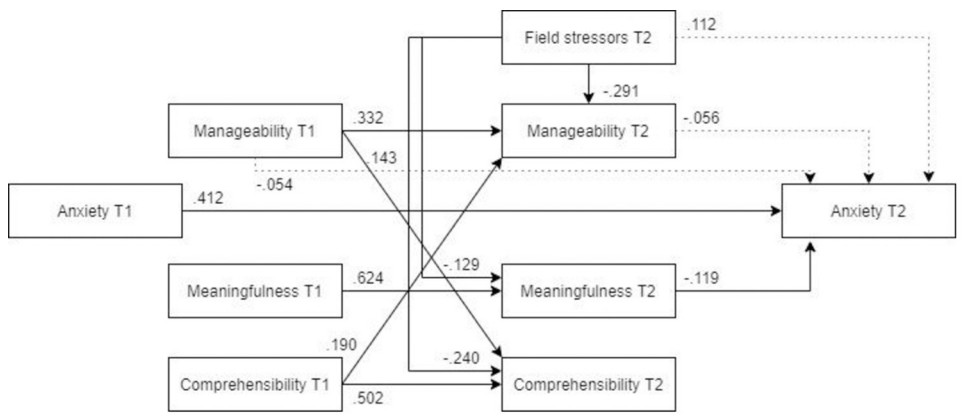

$$\chi2(9) = 23.084, \text{RMSEA} = .095 \ \text{CFI} = .972, \text{SRMR} = .061$$

**Fig 4. Tested structural equation model for anxiety—males.** $\chi2(9) = 23.084$, RMSEA = .095 CFI = .972, SRMR = .061.

predictor of post-assignment anxiety. Furthermore, field stressors were not significantly directly or indirectly associated with post-assignment anxiety.

*Females*. The pathways specified in the optimized model explained the relationships between variables significantly better for the female subsample than for the male subsample (Fig 5, N = 280). Compared to the optimized model for the whole sample, the relationship between post-assignment meaningfulness and post-assignment anxiety was not significant, and neither was the relationship between field stress and post-assignment meaningfulness.

**Depression.** *Model*. The hypothesized mediation model (Fig 6, N = 453) did not show an acceptable fit ($\chi2(12) = 310.10$, RMSEA .234, CFI .775, SRMR .118). Post-assignment depression was predicted by its pre-assignment value, and higher post-assignment SOC component scores were associated with lower post-assignment depression scores. A higher pre-assignment

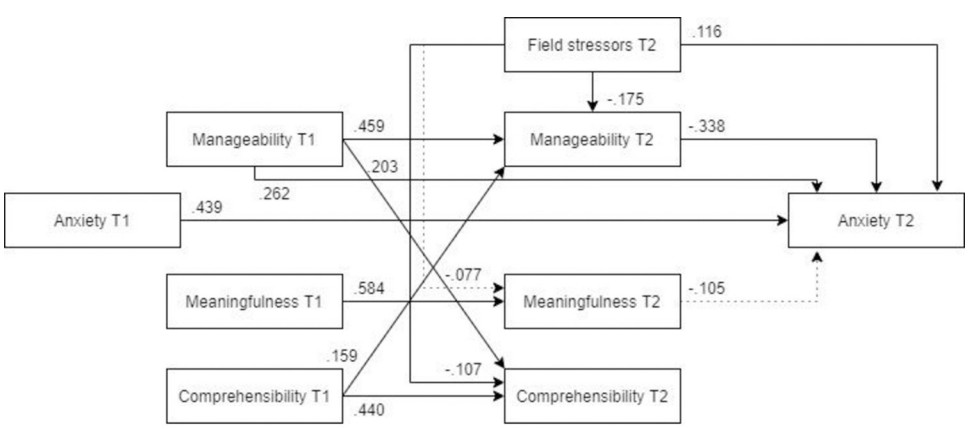

$$\chi2(9) = 13.083, \text{RMSEA} = .04, \text{CFI} = .994, \text{SRMR} = .036$$

**Fig 5. Tested structural equation model for anxiety—females.** $\chi2(9) = 13.083$, RMSEA = .04, CFI = .994, SRMR = .036.

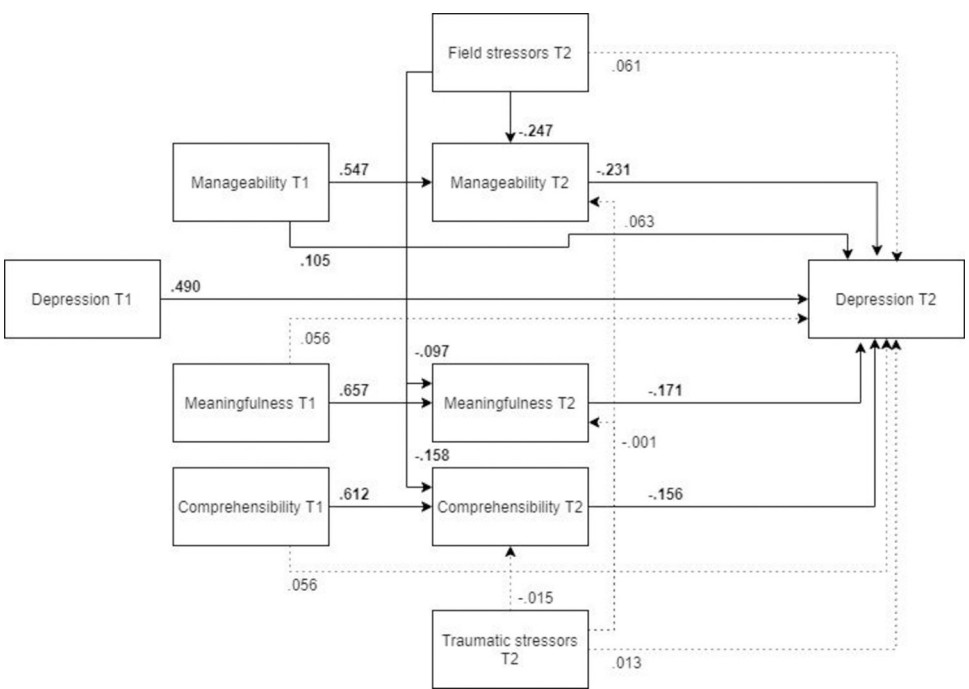

**Fig 6. Tested structural equation model for depression.**

manageability score, on the other hand, was associated with a higher post-assignment depression score. Neither field stressors nor traumatic stressors were directly predictive of post-assignment depression, but field stressors had an indirect detrimental impact.

*Optimized model.* The model was optimized by taking the same steps as earlier outlined for anxiety. The optimized model was a significantly better representation of the data than the hypothesized model (Fig 7, N = 453). All the remaining pathways were significant, and the final optimized model (MLFB) showed an adequate fit with the data ($\chi2(10)$ = 27.695, RMSEA = .063, CFI = .987, SRMR = .047).

*Male and female models.* **Males.** The optimized model for males (Fig 8, N = 173) showed a somewhat worse fit than for females, and was not an adequate representation of the

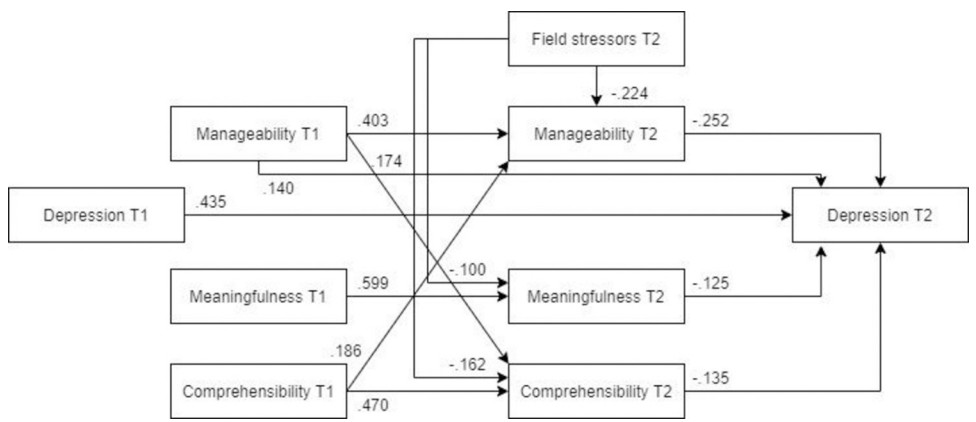

**Fig 7. Optimized structural equation model for depression.**

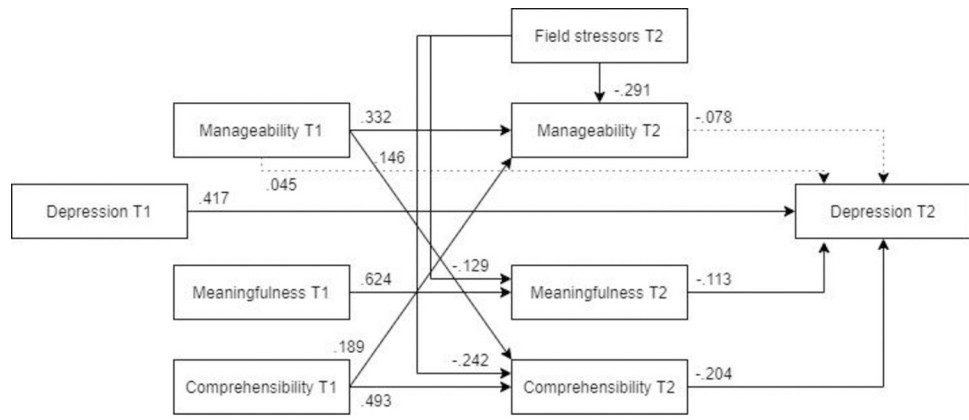

$$\chi2(10) = 21.189, RMSEA = .080, CFI = .978, SRMR = .06$$

**Fig 8. Tested structural equation model for depression—males.** $\chi2(10) = 21.189$, RMSEA = .080, CFI = .978, SRMR = .06.

relationships between the variables. Unlike in the optimized model for the whole sample, pre- and post-assignment manageability were not significant predictors of post-assignment depression.

*Females*. The optimized model showed a more adequate fit with the female subsample (Fig 9, N = 280) than the male subsample. Unlike for males and the overall sample, post-assignment comprehensibility was not a significant predictor of post-assignment depression. However, also opposed to the findings for males, post-assignment manageability had a strong beneficial association to post-assignment depression symptoms. Similar to the anxiety model, field stressors did not have a significant relationship with post-assignment meaningfulness.

**Emotional exhaustion.** *Model*. The model fit was not acceptable ($\chi2(12) = 320.33$, RMSEA .235, CFI .789, SRMR .120). As seen in Fig 10 (N = 465), post-assignment emotional exhaustion was directly predicted by its pre-assignment value, but also by field stressors and

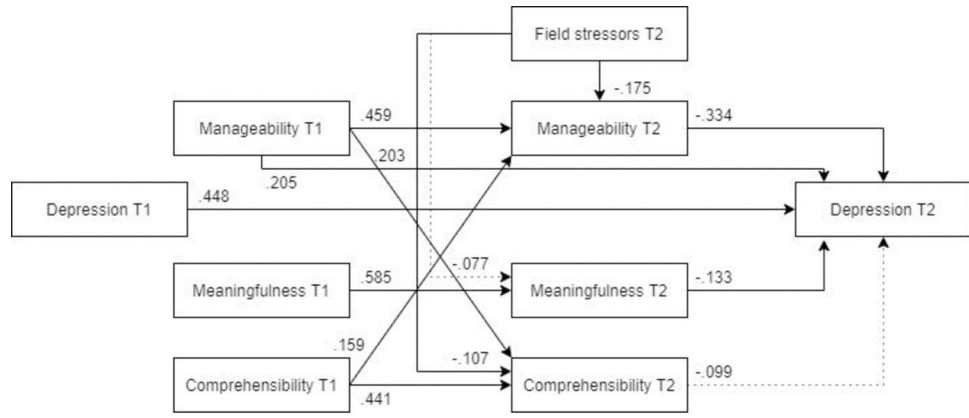

$$\chi2(10) = 14.254, RMSEA = .039, CFI = .995, SRMR = .04$$

**Fig 9. Tested structural equation model for depression—females.** $\chi2(10) = 14.254$, RMSEA = .039, CFI = .995, SRMR = .04.

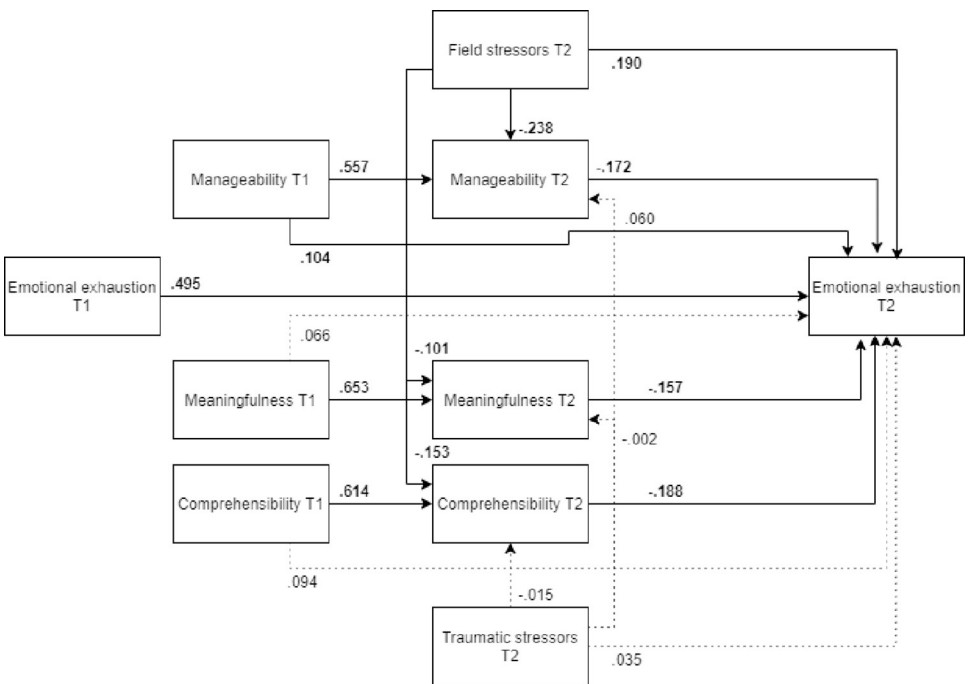

**Fig 10. Tested structural equation model for emotional exhaustion.**

pre-assignment manageability, both of which had a detrimental impact on post-assignment emotional exhaustion. All the post-assignment SOC components had significant beneficial associations with post-assignment emotional exhaustion. In addition to the direct detrimental effect, field stressors also had an indirect effect detrimental on post-assignment emotional exhaustion.

*Optimized model.* The model was optimized in the same manner as the earlier optimized models. This optimized model was a significantly better representation of the data than the hypothesized model (Fig 11, N = 465). Unlike in the earlier model, pre-assignment comprehensibility was also a significant predictor of post-assignment emotional exhaustion, showing a detrimental association with the outcome variable. All the remaining pathways were significant, but the final optimized model (MLFB) did not show an adequate fit with the data ($\chi2(8)$ = 36.121, RMSEA = .087, CFI = .981, SRMR = .054).

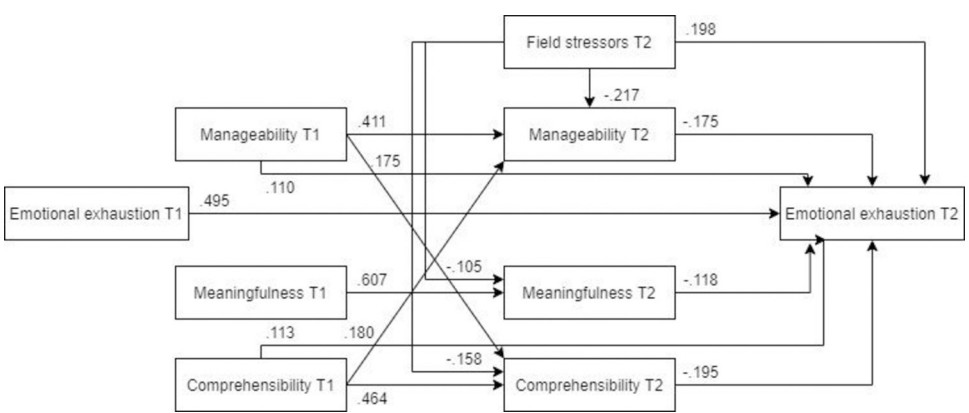

**Fig 11. Optimized structural equation model for emotional exhaustion.**

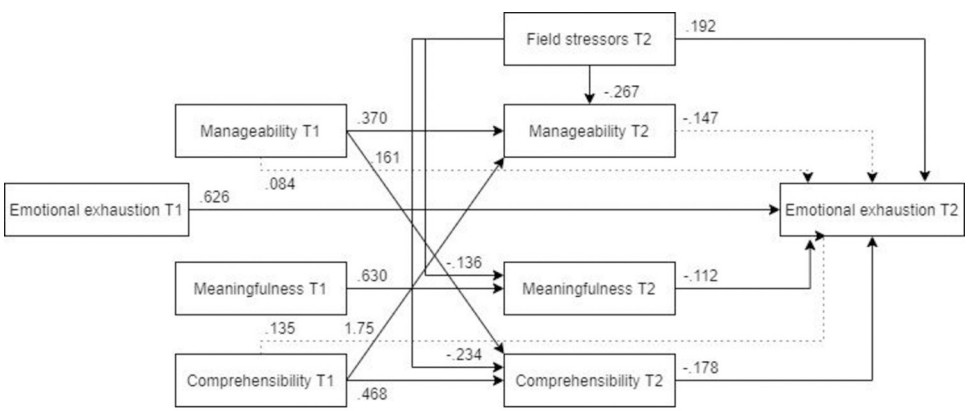

$$\chi 2(8) = 27.717, RMSEA = .117, CFI = .967, SRMR = .067$$

**Fig 12. Tested structural equation model for emotional exhaustion—males.** $\chi 2(8) = 27.717$, RMSEA = .117, CFI = .967, SRMR = .067.

*Model fit for males and females.* **Males.** The optimized model fit the male subsample (Fig 12, N = 180) significantly worse than it did the female subsample. Unlike in the overall optimized model, pre-assignment comprehensibility was not a significant predictor of post-assignment emotional exhaustion. Furthermore, the paths from pre- and post-assignment manageability to post-assignment emotional exhaustion were not significant.

*Females.* For females (Fig 13, N = 285), the optimized model showed a better fit than for males. Unlike in the optimized model for the full sample, the path from pre-assignment comprehensibility to post-assignment emotional exhaustion was not significant. Similar to the previous outcome variables, field stressors did not predict post-assignment meaningfulness.

**PTSD.** *Model.* The model fit was not acceptable ($\chi 2(12) = 308.43$, RMSEA .232, CFI .751, SRMR .113). Post-assignment PTSD symptoms were predicted by pre-assignment PTSD symptoms, as well as field stressors and post-assignment manageability (Fig 14, N = 460).

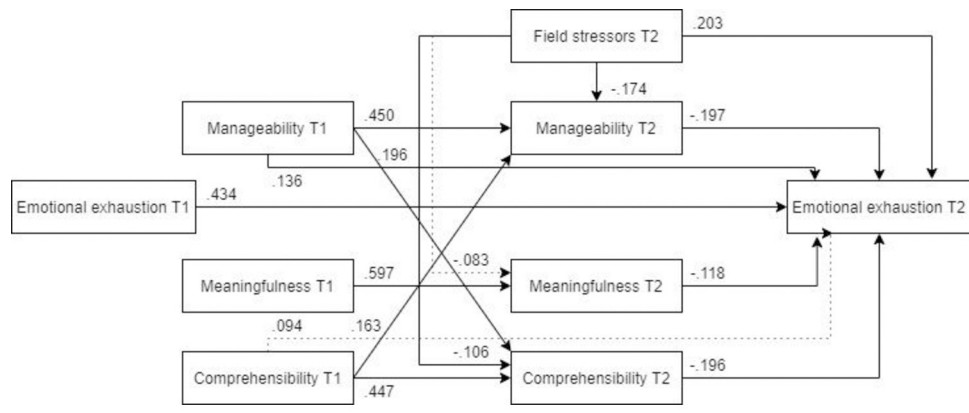

$$\chi 2(8) = 16.932, RMSEA = .063, CFI = .989, SRMR = .047$$

**Fig 13. Tested structural equation model for emotional exhaustion—females.** $\chi 2(8) = 16.932$, RMSEA = .063, CFI = .989, SRMR = .047.

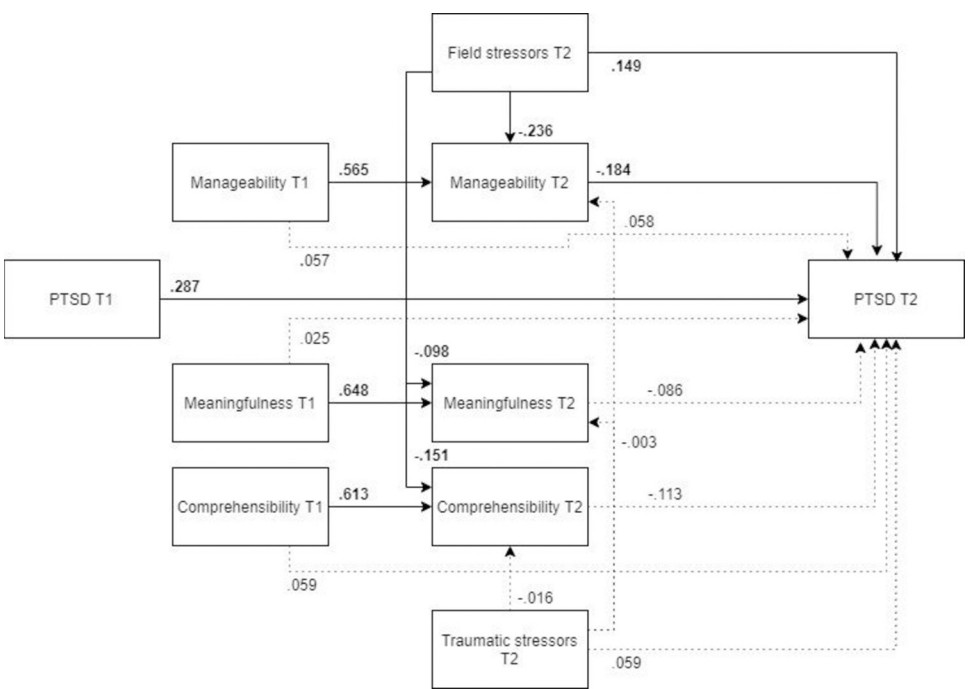

**Fig 14. Tested structural equation model for PTSD.**

Furthermore, field stressors had a further indirect effect on post-assignment PTSD symptoms. Traumatic stressors during the assignment had no impact on the post-assignment PTSD symptoms.

*Optimized model.* The model was optimized in the same manner as the earlier optimized models in this manuscript. The optimized model was a significantly better representation of the data than the hypothesized model (Fig 15, N = 460). All the remaining pathways were significant and the final optimized model (MLBF) showed an adequate fit with the data ($\chi^2(10)$ = 23.933, RMSEA = .055, CFI = .988, SRMR = .043).

*Male and female models.* ***Males.*** The MLBF model did not show as good a fit for males (Fig 16, N = 178) as it did for females. However, all the paths specified in the MLBF model remained significant.

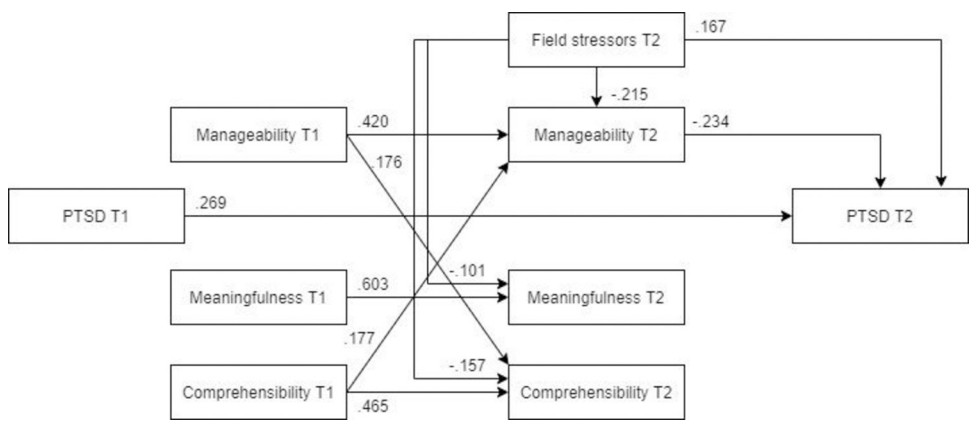

**Fig 15. Optimized structural equation model for PTSD.**

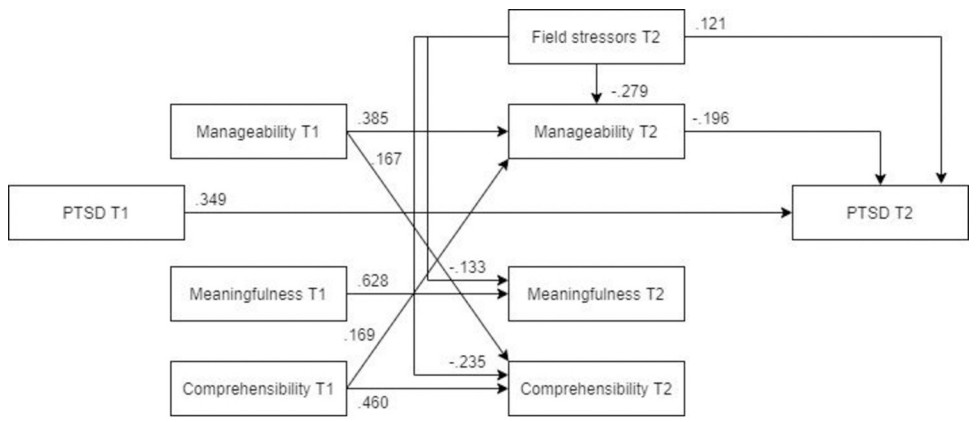

$$\chi2(10) = 23.103, RMSEA = .086, CFI = .974, SRMR = .065$$

**Fig 16. Tested structural equation model for PTSD—males.** χ2(10) = 23.103, RMSEA = .086, CFI = .974, SRMR = .065.

*Females*. The MLBF model showed a good fit for females (Fig 17, N = 282). Similar to the previous models, the relationship between field stress and meaningfulness was not significant; however, all other paths remained significant. Compared to males, field stressors had a larger detrimental impact on post-assignment PTSD in females, almost equaling that of pre-assignment PTSD.

**Work engagement.** *Model*. The model fit was not acceptable (χ2(12) = 330.97, RMSEA .244, CFI .768, SRMR .115). As it can be seen in Fig 18 (N = 446), post-assignment work engagement was significantly predicted by field stressors, post-assignment meaningfulness and pre-assignment comprehensibility, as well as pre-assignment work engagement. There was a small but significant indirect detrimental effect of field stressors on post-assignment work engagement.

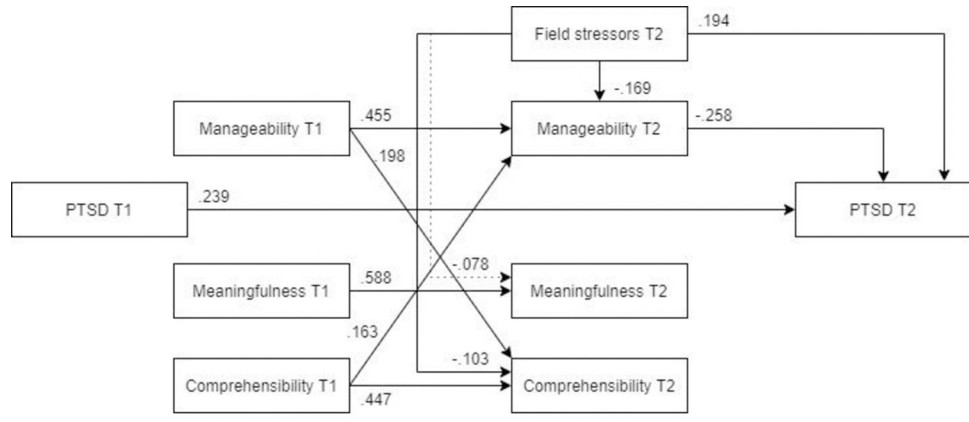

$$\chi2(10) = 11.016, RMSEA = .019, CFI = .999, SRMR = .037$$

**Fig 17. Tested structural equation model for PTSD—females.** χ2(10) = 11.016, RMSEA = .019, CFI = .999, SRMR = .037.

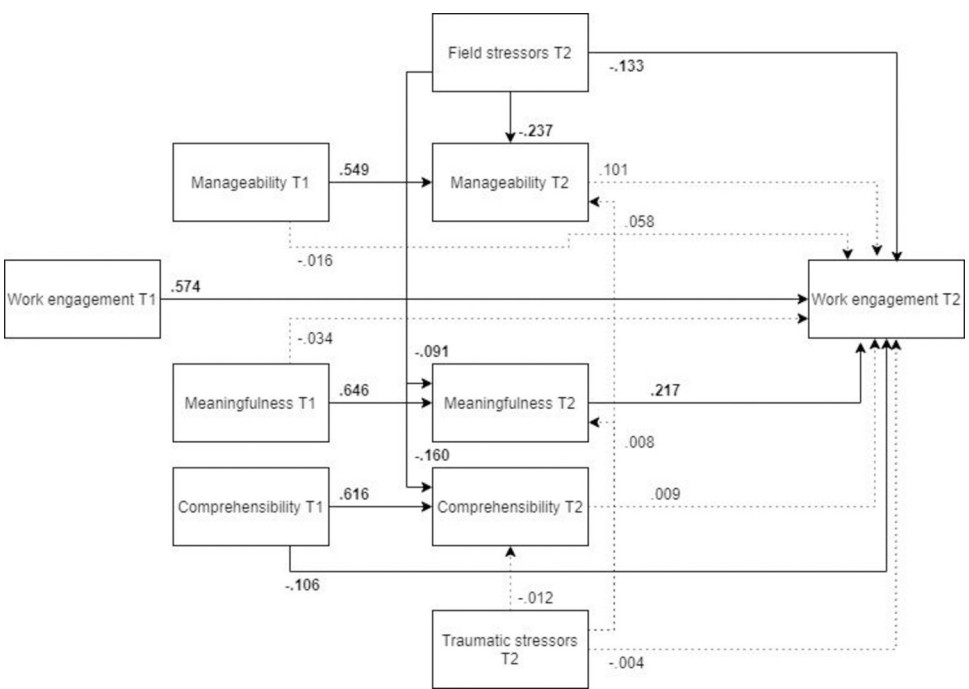

**Fig 18. Tested structural equation model of work engagement.**

*Optimized model* The model was optimized according to the same steps specified in the anxiety model. The optimized, MLBF model was a significantly better representation of the data than the hypothesized model (Fig 19, N = 446). Unlike in the original model, in the MLBF model post-assignment manageability was a significant predictor of post-assignment work engagement. All the remaining pathways were significant, yet the final MLBF model did not show an adequate fit with the data ($\chi$2(9) = 52.289, RMSEA = .104, CFI = .969, SRMR = .054).

*Male and female models*. **Males.** The MLBF model tested in males (Fig 20, N = 172) showed a less adequate fit than in the female subsample. Unlike in the MLBF model, the path from post-assignment manageability to post-mission work engagement was not significant, and

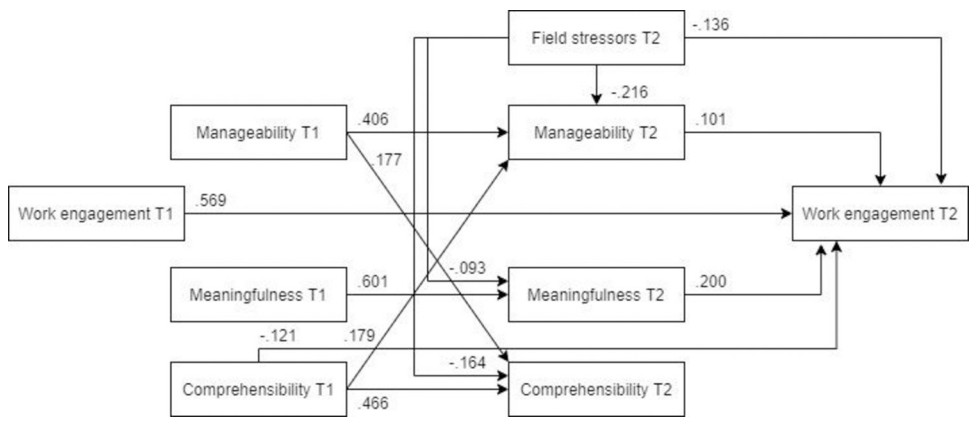

**Fig 19. Optimized structural equation model of work engagement.**

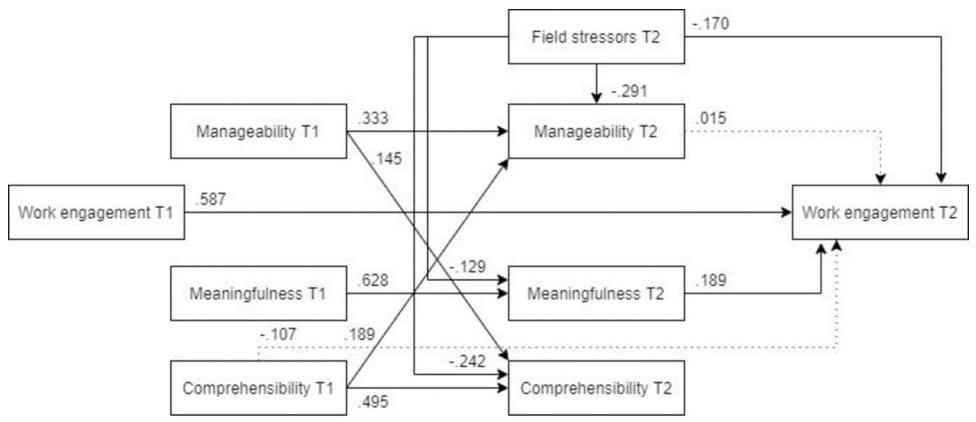

$$\chi2(9) = 36.063, RMSEA = .132, CFI = .952, SRMR = .066$$

**Fig 20. Tested structural equation model of work engagement–males.** χ2(9) = 36.063, RMSEA = .132, CFI = .952, SRMR = .066.

neither was the path from pre-assignment comprehensibility to post-assignment work engagement. Furthermore, field stressors had no indirect effect on post-mission work engagement.

*Females* The optimized model showed a better fit for females than for males (Fig 21, N = 274). Similar to the previous models, the relationship between field stress and meaningfulness was not significant. Other paths remained similar to those of the overall MLFB model.

## Discussion

The present study investigated whether SOC and its components mediated the relationship between traumatic and field stressors and health and work engagement in iHAWs. We also tested whether the SOC mechanisms differed between males and females. Key findings are discussed below.

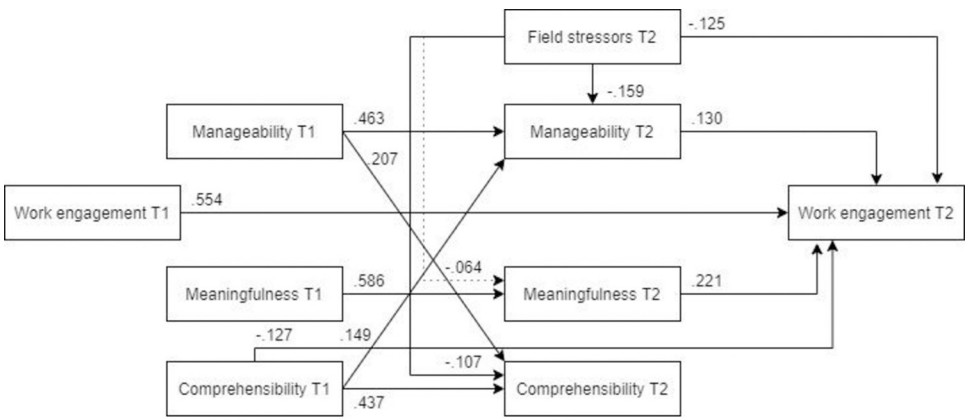

$$\chi2(9) = 30.740, RMSEA = .094, CFI = .973, SRMR = .048$$

**Fig 21. Tested structural equation model of work engagement–females.** χ2(9) = 30.740, RMSEA = .094, CFI = .973, SRMR = .048.

## The importance of being healthy before assignment

The strongest and most consistent predictor for staying healthy and engaged during assignment was good pre-assignment health and work-engagement. Health is a relatively stable state overtime despite the influence of external events and conditions [29].

## Traumatic stressors hardly affected health

Potentially traumatic stressors did not impact work engagement or any health indicators. There are two possible explanations for this notable finding. First, the labelling of a stressor as (potentially) traumatic is partly context-specific. People living in areas of mass conflict may register violence as a common occurrence, even if the same events are categorized as traumatizing in other contexts (e.g., in low-conflict societies) [30, 31]. IHAWs expect to endure hardship and exposure to shocking events due to the nature of their work. Therefore, they are more likely to appraise potentially traumatic experiences as part-of-the-job hardships, which they are trained to handle.

Second, a delayed onset response may also explain our finding. The demanding circumstances of an assignment do not allow for reflection time on one's health and experiences until one's return from the field. Only upon return, being in a quiet and safe environment, the trauma-related symptoms may emerge, or be recognized, and labelled as the consequence of a traumatic stressor. PTSD symptom increases during the first 2–6 months upon return from the field in both veterans and iHAWs support this assumption [1, 32].

## Field stressors had a detrimental impact on health

The difficult field-stressors [1] destabilise health and cause higher symptomatology on all health indicators and lower work engagement, both directly and indirectly. These findings follow the prevailing perspective that (occupational) stress causes health problems in iHAWs [33–35].

## SOC counteracted the detrimental impact of field stressors

Field stressors put a strain on the individual, causing health problems or lower work engagement. However, although field stressors have this negative effect, SOC mitigated some of it. In other words, SOC mediated the relationship between field stressors and health and work engagement. SOC, as reported before [10, 29, 30], had the ability to improve or maintain iHAWs' health and work engagement. How the different SOC components interplay specifically with the different health and engagement indicators is discussed below.

**SOC, depression and emotional exhaustion.** All three SOC components acted as a go-between for the impact of field stressors on post-assignment symptoms of depression and emotional exhaustion. The theoretical framework of SOC does not describe a sequential working of the different components. However, it is possible that the different components may work in sequence when dealing with depressive symptoms. Previous research has shown that the healing process of depression appears to start with the motivation to do something: the meaningfulness component was, at the onset of the depression treatment, the first predictor for successful healing [36]. Later on, understanding how the negative mood can be affected (comprehensibility) and what to do (manageability) to avoid or to deal with the negative affect of emotional conflict and uncertainty became the most important predictors for symptom improvement [36].

**SOC, anxiety and posttraumatic stress.** In the present study, the post-assignment SOC component manageability acted as a go-between for the negative effects of field stressors on

anxiety and posttraumatic stress responses. The significance of manageability reflected the importance of being confident to resolve anxiety and post-trauma effectively. A similar mechanism is postulated in general stress management theories in which effective coping with stress (e.g. feeling safe) depends on the confidence of personal and organizational resources being in place [37]. Firm meta-analytic evidence of a substantial negative correlation between SOC and PTSD supports our findings [38].

**The importance of 'doing good': Meaningfulness.** The post-assignment SOC meaningfulness component acted as a go-between for field stressors and work engagement, depression, and emotional exhaustion. Also, the negative impact of field stressors on post-assignment meaningfulness was substantially less compared to the impact of these stressors on comprehensibility and manageability; in the case of female iHAWs, meaningfulness was not even significantly impacted by field stressors. To do good and meaningful work appears to be an important driver of staying healthy and engaged [12]. Meaningfulness motivates iHAWs to mobilize the necessary resources to manage the extreme humanitarian field stress and to stay healthy and engaged. Meaningfulness is indeed a positive predictor of work engagement, which is associated with fewer health issues, such as depression [39]. When 'giving meaning' to one's actions fails, it may result in a negative cascade on the other SOC components and result in poor mental health. Meaningfulness defines the iHAWs community and may be a population-specific characteristic considering that those that perform aid work are likely self-selected based on the need to help others.

## Males and females used different SOC components to deal with the consequences of stress

A high sense of meaningfulness predicted better work engagement and health outcomes in both males and females, with the exception of PTSD. Also, in the case of females, meaningfulness was not a significant predictor of post-assignment anxiety. There were more sex differences in the use of manageability and comprehensibility in dealing with unescapable stressors (e.g., violence, being away from home). Males leaned towards their cognitive ability to clarify, understand and structure the nature of stressors (comprehensibility) to alleviate stress. Females tended to mobilise their available supportive resources (e.g., social support) to effectively deal with any negative consequences of the stressors via behavioural responses (manageability). Both strategies are effective and complementary. These different strategies have been described before among populations in distress, with females at risk of mental ill-health reporting greater difficulties in managing life compared to their healthy counterparts, and at-risk males reporting greater difficulties in comprehending one's life compared to their healthy counterparts [40].

## Conceptualizing SOC

The current findings also raise a number of conceptual discussion points.

**Context specific.** Unlike post-assignment SOC, pre-assignment SOC acted as a health-deteriorating mechanism. Higher pre-assignment SOC predicted post-assignment ill health (anxiety, depression, emotional exhaustion) and decreased work engagement. These findings were unexpected and contrast with prior findings of SOC as a predictor of good health and well-being [5].

The discrepancy between pre- and post-assignment SOC in promoting health may imply that SOC is not a stable *(fixed)* construct over time as viewed by Antonovksy [3]. The instability is likely attributable to retesting in a radically different environment. The pre-assignment SOC, with its Generalised Resistance Resources such as coping strategies, emotional closeness,

knowledge and intelligence, and support system [4], is not geared toward dealing with such new challenges, for example, arriving and working in a (new) humanitarian context. An environment that entails a different Generalised Resistance Resources (GRR), exerting their influence on SOC.

Transposing a pre-assignment SOC to function in a novel extreme stress environment may cause health issues if SOC is not adapted to the new environment. This may explain the pre-assignment SOC health-deteriorating mechanism described earlier. It may require iHAWs time to reassess current stressors, their meaning, available resources, to create a context-appropriate SOC to deal with new challenges.

Though SOC may be (more) stable in environments that do not change (much) over time [5] SOC, and its underlying GRR, may be best viewed as a context-specific or situational state-like disposition rather than a trait-like disposition or global life orientation [41, 42]. By accepting that SOC is not a stable construct we can explain the different response on the different 'environmental threats, challenges, demands, and resources (GRR). Based on our study findings we expect that future studies specify SOC as a flexible concept which will help to improve understanding of the different longitudinal trajectories. More importantly, it can help to specify guidance on supportive interventions, prevention measures, individual case management of those with deteriorated longitudinal trajectories.

**Multi-dimensionality.**   Antonovsky considered SOC to be a uni-dimensional construct. Nevertheless, the unidimensional conceptualization of SOC has been described as elusive [41, 43] and recent findings show SOC to be rather multi-dimensional [41]. The present study modelled each SOC component separately to advance this multi-dimensional conceptualization. The information our approach generated proved useful to understand how iHAWs stay healthy, such as how different SOC components act on different health indicators, the finding that meaningfulness appears less affected by field stressors compared to the other components, and the identification of different strategies used by males and females to cope with stress. These findings may extend current SOC theories regarding its mechanisms to promote health under stress.

## Implications

Leaving on assignment in good health is the most important predictor for staying healthy and engaged during field assignments. Pre-employment and pre-deployment mental health screening could be a useful way to detect staff suffering from health-related problems. However, currently, there is no research that shows such screenings are effective in predicting future disorders in staff [44]. Systematic advisory health conversations with iHAWs before and after assignment are a feasible alternative to improve the iHAWs' knowledge on how to stay healthy. These conversations should focus on current health status, knowledge of SOC as a context-specific personal health mechanism, and personal strategies to remain healthy.

Aid organizations that actively minimize the stress associated with humanitarian work will improve their employees' health and work engagement. Recommended is to focus on the most stressful field stressors [1]. For instance, by enhancing climate control, reducing dangerous and tiring outreach activities, and improving iHAWs' comprehension of the situational security and safety procedures. High-quality management, timely human resources replacement, technical support, good food, privacy, and internet connectivity help to further decrease levels of stress and exhaustion. Excessive workload is another major iHAW field stressor. It is important, despite contextual restrictions, to rationalise working hours, to create off-work recreational facilities and to organise social activities. Visser and colleagues [45] found out that especially trust in management was crucial for iHAWs to commit to a proper work-life

balance. In addition, iHAWs and management should be aware that field stressors are likely to exert a greater negative impact on health compared to potentially traumatic stressors for most iHAWs.

Strengthening SOC during aid assignments in a manner that best fits the appropriate field context is possible within a short space of time. General interventions that improve SOC and well-being [46] are, for example, physical workouts and mindfulness-based meditation practices, including mobile apps [47, 48]. A key SOC component in the process of staying healthy is meaningfulness. Ongoing communication on the purpose of the aid work, justification of choices and priorities, a culture of appreciation, and management actively seeking feedback from iHAWs [49] helps iHAWs to make sense of their work and may prevent mental health disorders [39].

Lastly, pre-departure SOC training, especially when it mimics the real humanitarian context, may strengthen the SOC during field assignments. Consider that males and females use different SOC components to stay healthy, a balanced sex composition is recommended in training and field team composition. It enables iHAWs to familiarise themselves with the different, complementary strategies of female and male iHAWs to mitigate inescapable stress.

## Strengths and limitations

The present study on iHAWs has several strengths. First, it was the first to examine the association between SOC and health or work engagement in iHAWs. Moreover, the longitudinal design and substantial sample size allowed us to use advanced statistical analyses for explaining how SOC and its components function as health mechanisms in iHAWs exposed to dangerous and demanding work environments. The inclusion of traumatic and work stressors separately allowed us to compare their differential impact on health. Finally, the use of SOC components allowed us to examine gender-specific approaches to maintain good health. The current health promoting findings also provide avenues of interest for other populations that operate in in similar settings, such as military personnel on deployment.

This study also has several limitations. First, while we found convincing evidence of mediating associations between SOC components and health outcomes, the current study design does not allow for causal inference. Second, the ratio of males and females was not equal in our sample. Therefore, when defining and testing the MLBF model, the sample with the largest N–in this case, females–determines most of the specifications of the MLBF, and that is likely why the MLBF is a better fit for females. Third, while obtaining a good model fit was not the focus of the current investigation, it can be noted that the model fit in some of the path analyses remained poor, indicating that they could not explain the (co)variance in our sample adequately. A poor fit can be attributed to testing specific theoretical pathways rather than allowing all model variables to co-vary with each other to better fit the data. More complex models could help to better fit the data and understand iHAW health changes, for example, by introducing general and specific resistance resources to the model. Both resource types facilitate the individual's abilities to cope with stressors (see [29], p.57-62, p.71-76). Fourth, the SOC component Cronbach's alpha scores were modest, indicating that a considerable proportion of the measurement scores may have been attributable to measurement error. This was not considered problematic because the reduction in alpha was expected due to the small number of items measuring each component [22], and the mean inter-item correlations demonstrated reliable component scores. Using the single latent SOC variable, with a high Cronbach's alpha score, delivered comparable model pathways and model fit results, and the results were generally in accordance with our expectations and prior findings.

## Conclusion

Being healthy and engaged before a humanitarian aid assignment is the best protection against high levels of stress that affect iHAWs' health during an aid assignment. The present study also demonstrated how the Sense of Coherence mediates the relationship between field stressors and health, enabling aid workers to remain healthy and engaged in their work. The different SOC components, especially meaningfulness, play an important role in this relationship. We also identified differences between male and female iHAWs in the SOC components. Our findings indicate that a context-specific perspective of SOC is appropriate. The implications of these findings are to be used by both organizations and professionals to promote better health of iHAWs before, during and after assignments, so that they can best serve those in need in war and disaster-affected areas.

## Supporting information

**S1 Appendix. Correlations between health outcome variables and predictor variables.**
(DOCX)

**S2 Appendix. Humanitarian field stressors list.**
(DOCX)

**S1 File.** A. Pre-modelling Step 1. B. Pre-modelling Step 2. C. Pre-modelling Step 3.
(ZIP)

## Author Contributions

**Conceptualization:** Kaz De Jong, Rolf Kleber, Joris Haagen, Ivan Komproe.

**Data curation:** Saara Martinmäki, Joris Haagen.

**Formal analysis:** Kaz De Jong, Saara Martinmäki, Rolf Kleber, Joris Haagen, Ivan Komproe.

**Funding acquisition:** Kaz De Jong.

**Investigation:** Saara Martinmäki, Joris Haagen.

**Methodology:** Kaz De Jong, Saara Martinmäki, Hans Te Brake, Rolf Kleber, Joris Haagen, Ivan Komproe.

**Project administration:** Kaz De Jong.

**Resources:** Kaz De Jong.

**Software:** Saara Martinmäki, Joris Haagen.

**Supervision:** Kaz De Jong, Hans Te Brake, Rolf Kleber, Ivan Komproe.

**Validation:** Kaz De Jong, Hans Te Brake.

**Writing – original draft:** Kaz De Jong.

**Writing – review & editing:** Saara Martinmäki, Hans Te Brake, Rolf Kleber, Joris Haagen, Ivan Komproe.

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
