## [Decision Letter · Decision Letter 0]

30 May 2022

PONE-D-21-37102How do international humanitarian aid workers stay healthy in the face of adversity?Is Sense of Coherence a mediator for health and work engagement?PLOS ONE

Dear Dr. de Jong,

Thank you for submitting your manuscript to PLOS ONE. After careful consideration, we feel that it has merit but does not fully meet PLOS ONE’s publication criteria as it currently stands. Therefore, we invite you to submit a revised version of the manuscript that addresses the points raised during the review process.  You have received two strongly positive reviews, but with a suggestion from Reviewer 2 that you are likely to want to consider in preparing a final version of your report. Specifically, the instability of sense of coherence scores may reflect shifts in their general resistance resources, such as social and emotional support.  Consideration of this fact may help your interpretation to be more consistent throughout the manuscript.  The changes can be minor but I wanted you to have the opportunity to make changes that you agree improve the manuscript.

Please submit your revised manuscript by August 15, 2022.  If you will need more time than this to complete your revisions, please reply to this message or contact the journal office at plosone@plos.org. Please include the following items when submitting your revised manuscript:A rebuttal letter that responds to each point raised by the academic editor and reviewer(s). You should upload this letter as a separate file labeled 'Response to Reviewers'.A marked-up copy of your manuscript that highlights changes made to the original version. You should upload this as a separate file labeled 'Revised Manuscript with Track Changes'.An unmarked version of your revised paper without tracked changes. You should upload this as a separate file labeled 'Manuscript'.If applicable, we recommend that you deposit your laboratory protocols in protocols.io to enhance the reproducibility of your results. Protocols.io assigns your protocol its own identifier (DOI) so that it can be cited independently in the future. For instructions see: https://journals.plos.org/plosone/s/submission-guidelines#loc-laboratory-protocols. Additionally, PLOS ONE offers an option for publishing peer-reviewed Lab Protocol articles, which describe protocols hosted on protocols.io. Read more information on sharing protocols at https://plos.org/protocols?utm_medium=editorial-email&utm_source=authorletters&utm_campaign=protocols.

We look forward to receiving your revised manuscript.

Kind regards,

C. Robert Cloninger, MD, PhD

Academic Editor

PLOS ONE

Journal Requirements:

3. Thank you for stating the following in the Competing Interests/Financial Disclosure* (delete as necessary) section:

“The first author may have competing interest as he is employed by Medecins Sans Frontiers. Both the funding of MSF as well as a clear description of the role of the financing body is mentioned in a separate statement in the manuscript. The role of the different authors in the research is described in detail in the manuscript.”

We note that one or more of the authors are employed by a commercial company: name of commercial company.

4. Please ensure that you refer to Figure 3, 5, 7 , 9, 11-21 in your text as, if accepted, production will need this reference to link the reader to the figure.

Reviewers' comments:

Reviewer's Responses to Questions

**Comments to the Author**

1. Is the manuscript technically sound, and do the data support the conclusions?

Reviewer #1: Yes

Reviewer #2: Yes

2. Has the statistical analysis been performed appropriately and rigorously? 

Reviewer #1: Yes

Reviewer #2: Yes

3. Have the authors made all data underlying the findings in their manuscript fully available?

Reviewer #1: No

Reviewer #2: Yes

4. Is the manuscript presented in an intelligible fashion and written in standard English?

Reviewer #1: Yes

Reviewer #2: Yes

5. Review Comments to the Author

Reviewer #1: Thank you for the opportunity to review this paper which is about International Humanitarian Aid Workers (iHAWs) and the fact that despite working under stressful and traumatic environments, they remain healthy. The authors used a stress-health model to test whether the concept of a Sense of Coherence (SOC) as a health mechanism would mediate the relationship between stressors related to the field work of iHAWs after they returned from their assignment.

I found this paper very interesting and very well written throughout. The background to the concept and components of SOC was clearly defined as was their rationale for choosing it to explore how and why iHAWs remain relatively healthy despite challenging work and conditions. Salutogenesis was sufficiently explained as the rationale for looking at the three components of SOC.

Methods were clear and I feel their choice of instruments was appropriate to provide the various indicators of health in their model.

Analyses seems comprehensive and appropriate. The number of participants is adequate for the analyses they performed. Although they noted some limitations to further analyses due to sample size. Although not an expert, overall, their statistical analysis performed, and their mediation analysis seems correct. They thoroughly tested the demographic variables, mainly sex, as they may impact on their findings.

The discussion again was clearly written and described the results while highlighting how certain variables were mediated by the three components of SOC. I found this very interesting and the I think they have strengthened the importance of considering the context in which any action and behaviour is observed. I feel this information could go beyond the promotion of better health for workers in difficult and challenging environments but can be applied more broadly to situations where the context is specifically unique and/or intense.

Reviewer #2: Dear authors, my only remark is regarding the last sentence on page 21. In it, you assume that the SOC may not be a stable construct over time, according to your results. I would like to emphasize that the SOC is a heterogeneous construct and this could lead to erroneous conclusions. Please keep in mind that this construct is identical to Generalized Resistance Resources (GRRs), which in practice remain undiscovered by the researchers who measure only the underlying trend for coherence, or SOC. Therefore, measuring pre- and post-assignment SOC in your study is not an indicator of construct instability. During this period, the participants in your study has experienced immeasurable dynamics of GRRs. In my opinion, this explains the discrepancy between pre- and post-assignment SOC in promoting health. I remind you that GGRs includes coping strategies, emotional closeness, knowledge and intelligence, and support system. Refer to: Mittelmark, M. B., Sagy, S., Eriksson, M., Bauer, G. F., Pelikan, J. M., Lindström, B., & Arild Espnes, G. (2017). The handbook of salutogenesis. Springer Nature.

In short, I urge you to rethink this sentence in your work. On page 24 of the manuscript, which describes the strengths and limitations, you yourself self-critically point out that the design does not allow for causal conclusions. That is enough !!!

I sincerely congratulate you on the idea of researching SOC in these samples, which are so presumptively focused on pathology!

6. PLOS authors have the option to publish the peer review history of their article (what does this mean?). If published, this will include your full peer review and any attached files.

Reviewer #1: **Yes: **Diann S Eley

Reviewer #2: No

---

## [Author Response · Author response to Decision Letter 0]

21 Aug 2022

We appreciate highly the comments of the reviewers and the editors.

---

## [Decision Letter · Decision Letter 1]

13 Oct 2022

How do international humanitarian aid workers stay healthy in the face of adversity?

PONE-D-21-37102R1

Dear Dr. de Jong,

We’re pleased to inform you that your manuscript has been judged scientifically suitable for publication and will be formally accepted for publication once it meets all outstanding technical requirements.

Kind regards,

Silva Ibrahimi, PhD

Academic Editor

PLOS ONE

Additional Editor Comments (optional):

Reviewers' comments:

Reviewer's Responses to Questions

**Comments to the Author**

1. If the authors have adequately addressed your comments raised in a previous round of review and you feel that this manuscript is now acceptable for publication, you may indicate that here to bypass the “Comments to the Author” section, enter your conflict of interest statement in the “Confidential to Editor” section, and submit your "Accept" recommendation.

Reviewer #2: All comments have been addressed

Reviewer #3: All comments have been addressed

2. Is the manuscript technically sound, and do the data support the conclusions?

Reviewer #2: Yes

Reviewer #3: Yes

3. Has the statistical analysis been performed appropriately and rigorously? 

Reviewer #2: Yes

Reviewer #3: Yes

4. Have the authors made all data underlying the findings in their manuscript fully available?

Reviewer #2: (No Response)

Reviewer #3: Yes

5. Is the manuscript presented in an intelligible fashion and written in standard English?

Reviewer #2: Yes

Reviewer #3: Yes

6. Review Comments to the Author

Reviewer #2: I agree with the positions the authors defend regarding my criticisms. I accept that the text and the research are sufficiently well-argued.

Reviewer #3: This is a highly significant argument and interesting research. It provides both an empirical and broad statistical view on such a issue as the humanitarian work and wellbeing. I would kindly suggest authors only to replace the word "analyses" to "analysis".

7. PLOS authors have the option to publish the peer review history of their article (what does this mean?). If published, this will include your full peer review and any attached files.

Reviewer #2: No

Reviewer #3: No

---

## [Editor Report · Acceptance letter]

24 Oct 2022

PONE-D-21-37102R1 

How do international humanitarian aid workers stay healthy in the face of adversity? 

Dear Dr. de Jong:

I'm pleased to inform you that your manuscript has been deemed suitable for publication in PLOS ONE. Congratulations! Your manuscript is now with our production department. 

Kind regards, 

on behalf of

Dr. Silva Ibrahimi 

Academic Editor

PLOS ONE